# Innate Immune System Response to Burn Damage—Focus on Cytokine Alteration

**DOI:** 10.3390/ijms23020716

**Published:** 2022-01-10

**Authors:** Olga Sierawska, Paulina Małkowska, Cansel Taskin, Rafał Hrynkiewicz, Paulina Mertowska, Ewelina Grywalska, Tomasz Korzeniowski, Kamil Torres, Agnieszka Surowiecka, Paulina Niedźwiedzka-Rystwej, Jerzy Strużyna

**Affiliations:** 1Doctoral School of the University of Szczecin, Institute of Biology, University of Szczecin, 71-412 Szczecin, Poland; olga.sierawska@phd.usz.edu.pl (O.S.); paulina.malkowska@phd.usz.edu.pl (P.M.); 2Biology Department, Faculty of Science, Ankara University, Ankara 06560, Turkey; canseltaskinn@hotmail.com; 3Institute of Biology, University of Szczecin, 71-412 Szczecin, Poland; rafal.hrynkiewicz@usz.edu.pl; 4Department of Experimental Immunology, Medical University of Lublin, 20-093 Lublin, Poland; paulinamertowska@umlub.pl (P.M.); ewelina.grywalska@gmail.com (E.G.); 5Chair and Department of Didactics and Medical Simulation, Medical University of Lublin, 20-093 Lublin, Poland; tomasz.korzeniowski@umlub.pl (T.K.); kamil.torres@umlub.pl (K.T.); 6East Center of Burns Treatment and Reconstructive Surgery, ul. Krasnystawska 52, 21-010 Łęczna, Poland; dr.surowiecka@gmail.com (A.S.); jerzy.struzyna@umlub.pl (J.S.); 7Chair and Department of Plastic, Reconstructive Surgery and Burn Treatment, Medical University of Lublin, 20-093 Lublin, Poland

**Keywords:** immune system, burns, microorganism

## Abstract

In the literature, burns are understood as traumatic events accompanied by increased morbidity and mortality among affected patients. Their characteristic feature is the formation of swelling and redness at the site of the burn, which indicates the development of inflammation. This reaction is not only important in the healing process of wounds but is also responsible for stimulating the patient’s innate immune system. As a result of the loss of the protective ability of the epidermis, microbes which include bacteria, fungi, and viruses have easier access to the system, which can result in infections. However, the patient is still able to overcome the infections that occur through a cascade of cytokines and growth factors stimulated by inflammation. Long-term inflammation also has negative consequences for the body, which may result in multi-organ failure or lead to fibrosis and scarring of the skin. The innate immune response to burns is not only immediate, but also severe and prolonged, and some people with burn shock may also experience immunosuppression accompanied by an increased susceptibility to fatal infections. This immunosuppression includes apoptosis-induced lymphopenia, decreased interleukin 2 (IL-2) secretion, neutrophil storm, impaired phagocytosis, and decreased monocyte human leukocyte antigen-DR. This is why it is important to understand how the immune system works in people with burns and during infections of wounds by microorganisms. The aim of this study was to characterize the molecular pathways of cell signaling of the immune system of people affected by burns, taking into account the role of microbial infections.

## 1. Introduction

The skin is the largest organ of the body and constitutes about 16% of a person’s weight. It consists of two layers—a thinner outer layer, i.e., the epidermis, and a deeper, thicker layer, i.e., the dermis. Various other structures are found in the skin, such as hair follicles, sebaceous glands, sweat glands, capillaries, and nerve endings. The main role of the skin is to protect against environmental aggression, such as infections, temperature changes, physical forces, chemicals, etc. In addition, it has immunological, neurosensory and metabolic functions, as well as water homeostasis and thermoregulation. 

A burn is considered to be damage to the skin due to the action of heat, corrosive chemicals, electric current, solar UV rays, radiation (X-ray, UV and other extreme radiation factors), and depending on the degree, the burn can reach deeper tissues or organs [1,2,3]. The most common are thermal burns, followed by electrical and chemical burns. The most common victims of burns are patients aged 15-64 (>60% of cases) and children up to 4 years of age (20% of cases) [1,3,4].

Burns are a serious problem. In the United Kingdom, around 250,000 people are burned each year, and there are 300 deaths as a result. Data from the United Kingdom are representative of most developed countries, although some, such as the United States, have a higher incidence of burns [4]. In turn, India experiences over two million burns annually, and this may be a significant underestimation [4]. According to the World Health Organization, there are approximately 300,000 burn deaths annually [2], and according to the American Burn Association data, in America in 2012, the total annual number of burn-related deaths was approximately 3400 [5]. Burns are the third leading cause of death in children aged 5 to 9 [6]. The main cause of death after the first 24 hours is sepsis and the accompanying invasive infection [1,5,7], which is caused by pathogens penetrating the damaged skin that is deprived of its protective function [2,8].

Burns cause several systemic responses: (a) increase in the permeability of capillaries, (b) occurrence of bronchospasms, (c) reduction of the contractility of heart muscle, (d) contraction of the peripheral and visceral vessels, and (e) increase in metabolism up to three times the original rate. There is non-specific downregulation of the immune response, which affects cellular and humoral pathways (Figure 1) [1,4]. The immune response initiates both proinflammatory and anti-inflammatory phases, simultaneously or sequentially, to maintain homeostasis and normal physiology [3]. Thermal damage causes 80% of leukocyte transcriptome to change, leading to stimulation of innate genes (both proinflammatory and anti-inflammatory) and the suppression of adaptive immune responses [9]. Neutrophil dysfunction, release of immature granulocytes, and a decreased number and disturbed expression of CD14+/HLA-DR+ monocytes have also been observed [10,11]. Conversely, up to 3 years after the burn, there may be a simultaneous increase in the level of granulocyte–macrophage colony stimulating factor (GM-CSF), interleukin 10 (IL-10) and other cytokines [12].

We can distinguish three zones in burns, which were described by Jackson (Figure 2) [13]. The outermost zone is the zone of hyperemia, where tissue perfusion is increased. Tissue here will return to normal over time unless severe sepsis or prolonged hypoperfusion occurs. The middle zone is the zone of ischemia (or stasis), where there is reduced tissue perfusion. This area has an increased risk of progressing to necrosis due to hypoperfusion or infection. The inner zone is the zone of coagulation (or necrosis), which occurs at the point of maximum damage and is the area of irreversible cell death because of the coagulation of proteins [1,4]. The direction of trauma in burns is usually horizontal, and although it seems that the burn wound should be free of microorganisms because of the high temperature, rapid blisters and the necrosis of damaged tissue open the wound to pathogens, which is associated with the risk of infection [14,15]. Infections in both adults and children with burns include pneumonia, cellulitis and urinary tract infections [16].

In this work, we will review the changes that occur in the immune system of a person who undergoes burns, based on the latest literature data.

## 2. Host Immune Response to Burns

At the site of the burn wound, loss of primary tissue is observed as a result of protein denaturation, hence a release into the circulation of toxic proinflammatory mediators and platelet activating factors. Immune cells, similar to macrophages and neutrophils, produce oxygen free radicals (ROS) to destroy pathogens, which damage skin structures, and which results in a strong inflammatory response in the immune system, defined as systemic inflammatory response syndrome (SIRS) [3]. Innate immunity provides non-specific and rapid responses to infection [17], and its mechanisms are the first to respond and recognize microbial pathogen-related molecular patterns (PAMPs) and danger-related molecular patterns (DAMPs). This is achieved using the instrumentation of pattern recognition receptors (PRRs), which include several classes: Toll-like receptors (TLR), NOD-like receptors (NLR), RIG-I-like receptors (RLR) and C-type lectin receptors (CLR) [2,18]. Outside of the body, epithelial cells secrete defensins, lysozymes and cathelicidins, whose roles are to destroy the cell walls of pathogens and provide antibacterial activity [17]. Inside the body, the innate immune system consists of macrophages, neutrophils, dendritic cells (DC) and more. They provide a rapid and non-specific response to commonly known antigens. The activation of these pre-programmed cells occurs when these fragments bind to TLRs on the surface of these immune cells. TLRs also detect common antigenic molecules such as lipoteichoic acids from Gram-positive bacteria and lipopolysaccharides/lipoproteins from Gram-negative bacteria. Binding of PAMP to TLRs leads to rapid phagocytosis of suspect pathogens. In addition, innate immunity involves the recruitment of non-cellular defense mechanisms such as complement, coagulation and proinflammatory cytokine proteins that enhance the acquired immune system [17].

The first cells that respond to burns are mast cells, neutrophils, DCs and monocytes, which migrate to the site of inflammation, initiated by chemotactic factors released from the coagulation process (kallikreins, fibrin peptides) and substances released from mast cells (tumor necrosis factor, histamine, proteases, leukotrienes and cytokines). The cellular response supports phagocytosis and the clearance of necrotic tissue and toxins being released by tissues that have been burned [3,19,20]. This is critical because the burn injury causes a disruption of the mononuclear phagocyte system (MPS), resulting in a disruption of phagocytosis [21]. In addition to phagocytosis impairment, macrophages also exhibit abnormal intracellular killing actions and chemotaxis [19]. 

### 2.1. Mast Cells

Mast cells are analogs of basophils in the blood. Mast cells mature in connective tissue and mucous membranes and remain in various tissues for several weeks. The differentiation of these cells occurs depending on the stem cells factor (SCF), also known as the mast cells growth factor (MGF) [22].

Burn injury leads to an increase in the number of mast cells in tissues and subsequently stimulates their degranulation and release of histamine, heparin, and enzymes such as chymase, cathepsin G, and hydroxypeptidase A [2,14,23,24]. The substances secreted by MCs promote wound healing, including those caused by burns [25]. Burn injury causes MCs’ atrophy, and subsequently, high levels of MCs are observed in burn scar tissue, consistent with their function in promoting wound healing [26] by up to 10–100 times higher than in healthy skin. In rodent models, high levels of cells persist up to 30 days after burn [27].

### 2.2. Neutrophils 

In addition to NK and NKT cells, myeloid cells (macrophages, neutrophils, and DCs) that express surface receptors and that activate when stimulated by PAMPs are heavily involved in the immediate immune response [2]. As a result of the immune response, neutrophils produce MMPs, collagenases and elastases, and secrete proinflammatory cytokines. To remove pathogens, neutrophils will use phagolysosomes, free radical release, and antimicrobial proteases that damage cell membranes and trap microbes in histone and DNA networks [14]. Neutrophils can create and release extracellular neutrophil traps (NETs), which are spatial structures made of chromatin fibers with histones and granulocyte granular proteins attached, such as: lysosomal proteases, antibacterial peptides, or proteins responsible for the oxygen inactivation of microorganisms. The structure and biochemical composition of NETs enable efficient trapping and then rapid elimination of pathogens [28].

Neutrophils rapidly infiltrate burn wounds. Hampson et al. [10] reported that, within 24 hours after the injury, the number of neutrophils increases significantly, and the state normalizes 3 days after the injury; then, it increases significantly on day 7 and remains elevated for 28 days after the burn [10]. Neutrophils produce oxidants, antimicrobial peptides and proteolytic enzymes and provide non-specific antimicrobial defense [17]. After the burn, the ability of neutrophils’ oxidative burst decreases, which in turn weakens phagocytosis [10,29]; there is a decrease in the speed of motility during chemotaxis [30,31] and decreased bactericidal capacity [14], which ultimately reduce innate immune function. Hampson et al. [10] reported a reduced ex vivo NET generation (followed by a death process—NETosis) in patients with major burns [10]. Jones et al. [32] described a new phenotype of neutrophil migration—spontaneous migration in the absence of chemoattractants that are strongly correlated with the occurrence of sepsis in the burns of patients. These changes were observed 2 days before the development of sepsis, which may prove helpful in post-burn treatment because identification of infections in the blood requires at least 12–24 hours of blood culture before the bacteria reach levels that can be detected [32].

Typically, burns adversely affect neutrophil function by impairing their ability to phagocytize, engage in chemotaxis, produce reactive oxygen species (ROS) and form neutrophil extracellular traps (NETs) [30]. However, this is hazardous, as the antimicrobial properties of neutrophils are primarily based on phagocytosis and the production of ROS and NETs [10]. Impaired chemotaxis in neutrophils is manifested primarily by a reduction in chemotactic distance and a decreased number of chemotactic cells, which results in the inability to reach damaged sites [31]. To prevent inflammation, which can damage tissues adjacent to the burn wound, neutrophil apoptosis is induced by macrophages, and then neutrophils are phagocytosed, and the macrophages involved in this process become apoptotic [33].

The high accumulation of neutrophils in various tissues during the early phase of the burn suggests that they may be a source of ROS. ROS production following a severe burn may also cause damage to distant organs [34]. The inflammation that occurs as a result of burn injury is also present in uninjured tissues and makes the body unable to manage on its own in maintaining the balance between ROS production and destruction, causing oxidative stress [35]. ROS produced by neutrophils are one of the mediators and occur in increased amounts in a burn injury [36].

### 2.3. Dendritic Cells

Dendritic cells (DC), also known as dendritic leukocytes (DL), are a family of phagocytes belonging to professional antigen presenting cells (APC). DCs migrate to the lymph nodes to present antigens to CD4 and CD8 T cells in order to initiate an acquired immune response [37,38]. CD103+ DCs present both CD4 and CD8 antigens, and CD11b+ DCs mainly present antigens to CD4 T cells [37].

Patients with burns showed a decrease in the amount of DC, inhibition of myeloid-derived DC (mDC) production [20,38], and a reduced ability to induce T-cell activation by LC and DC [38]. Valvis et al. [37] reported a significant increase in the DCs CD8a+ with a decreased activation state after burn. It has been shown that the presence of DC influences the early proliferation of cells occurring 2–3 days after the burn; when there is a loss of DC, it is significantly reduced [20].

### 2.4. Monocytes and Macrophages

Both monocytes and macrophages play an important role in the healing of burns by fostering the production of fibroblasts and keratinocytes [39]. Also, they are classified into classical activated (proinflammatory), non-classical activated (anti-inflammatory) and intermediate cells [40].

Monocytes are primarily responsible for phagocytosis, production of cytokines and antigen presentation [28]. Activated by the injury, monocytes initiate the production of chemokines and cytokines that attract other cells of the immune system, such as, e.g., neutrophils, to the site of injury and enhance their response [17,28,41]. Production of cytokines and chemokines are stimulated after monocytes recognize pathogen-associated molecular patterns (PAMPs) and damage-associated molecular patterns (DAMPs) [42]. Monocyte migration to the burn site is associated with specific chemoattractants, such as IL-6, IL-8 and IL-8β, adenosine and the lymphocyte function-associated antigen-1 (LFA-1) complex [14,43]. Burns increase the number of peripheral blood monocytes [44]. It has been shown that burn patients have fewer monocytes expressing the human leukocyte antigen (HLA-DR), and their number is more reduced in patients with burn sepsis [45,46]. Monocytes leave the bloodstream to reach endangered tissues and become tissue macrophages there [17,28,41].

Macrophages act as significant phagocytes, with the function of direct microbial killing, as do neutrophils. Macrophages include cutaneous (epithelial) Langerhans cells (LC). They are found in tissues exposed to foreign antigens, and when an infection occurs, they secrete cytokines that attract and activate other cells of innate immunity to enhance killing. Their primary function is to regulate T and B cell responses to specific antigens [17,47]. In macrophage populations, an initial increase in classical activated is followed by an increase in non-classical activated signaling in the later inflammatory phase [14,38]. Macrophage hyperactivity after burn is associated with changes in intracellular cAMP, the levels of which are increased in immune cells after thermal shock [48]. cAMP is produced by activation of cell surface receptors. They play a role as a second messenger in signal transduction pathways by activating cAMP-dependent kinases (PKA), which leads to the phosphorylation of intracellular proteins [48,49,50]. According to research conducted by Schwacha et al. [48], in burned animals, cAMP levels in macrophages are increased due to the changes that happen in downstream signaling of adenylate cyclase activation and cAMP. Macrophage hyperactivity is also related to a rise in ATP-consuming reactions, such as protein synthesis (22%), hepatic gluconeogenesis (10%) and cycling of glucose and fatty acids (21%). Overall, more than half of the hypermetabolic responses consist of ATP-consuming reactions [51,52].

### 2.5. Inflammasomes

Inflammasomes are multimeric protein complexes that accumulate in the cytosol upon detection of PAMP or DAMP and whose key role is to regulate caspase-1 activation [53]. They are multi-protein complexes composed mainly of protein receptors of the NLR or ALR family, with apoptosis-associated speck-like proteins containing a C-terminal caspase recruitment domain (CARD). The role of NLRP3, NLRC4 and AIM2 inflammasomes in infections and diseases of various backgrounds has been demonstrated [18].

NLRP3 expression is closely associated with patient mortality and has a significant role in the post-burn acute phase [54]. Severe thermal times cause not only inflammation but also metabolic changes. When these are not controlled, severe metabolic dysfunction can occur, during which the NLRP3 inflammasome is stimulated. NLRP3 causes activation of caspase-1, which converts pro-IL-1β to its mature form. IL-1β affects metabolic tissues and causes changes in insulin signaling and insulin resistance. These changes often cause stress-induced diabetes in patients with severe burns, which in turn contributes to increased mortality [55]. In addition, NP3R has the capability to identify DAMPs that are released after burn injury [2]. One hour after the burn, a deficiency in NLRP3 expression was observed, leading to a decrease in the expression of factors involved in wound healing processes. Reduced production of proinflammatory cytokines and chemokines impairs not only keratinocyte migration and proliferation, but also chemotaxis of immune cells [56]. It has been reported that NLRP3 is secreted by classically activated macrophages and has a significant role in activating the proliferative healing phase of burn wounds through the release of proinflammatory cytokines, chemokines and growth factors [57]. Conversely, studies by Xiao et al. [58] on rat models showed that NLRP3, secreted mainly by the zone of stasis’ macrophages, can negatively affect burn wound healing by producing too many cytokines [56]. Related results have been shown by Han et al. [22], where NLRP3 activity was enhanced at burn onset, and impairment of its activity attenuated the severity of burn-induced acute lung injury (ALI) [22]. Clearly, further studies are needed to determine the exact role of NLRP3 in burn patients.

### 2.6. NK and NKT Cells

NK and NKT cells release cytotoxic granules that attach to infected cells and induce apoptosis. NK cells are activated by type I interferons, and when activated, they induce the synthesis of type II interferons, interferon-γ (IFN-γ), tumor necrosis factor-α (TNF-α) [2], perforin and granzymes [17] and present the antigen [59]. They express various receptors, including: NCRs (natural cytotoxicity receptors), belonging to the immunoglobulin family of natural cytotoxicity receptors; KIRs (killer cell immunoglobulin-like receptors), immunoglobulin-like killer cell receptors that interact with class I major histocompatibility complex (MHC) molecules on the surface of NK cells during the transmission of signals that activate or inhibit the activity of these cells; lectin-like receptors, involved in the activation or inhibition of NK cell lectin-like receptors; NK cell activating receptors (including NCR, KAR, NKG2D, -C and -E and Ly49D, -H, -P and -W) and NK cell activity inhibitory receptors (Ly49A, -B, -C, -E, -F, -G, Ly491, NKG2A, -B and IL-T2) [59,60]. NCRs include several major receptors, such as Natural Killer Group 2D (NKG2D), which is not limited to NK cell expression and mediates the recognition of damaged, transformed cells, and Pseudomonas clearance [59].

NK cells, due to their ability to kill without recognizing the histocompatibility complex, constitute one of the first lines of defense against viral infections associated with increased mortality and morbidity after severe thermal injuries [28]. A reduced ability to fight Herpes Simplex-1 virus, the most common viral infection affecting burn patients, has been observed [17,61].

NK cell deficiencies are greatest after the first week after burn injury, possibly related to the impairment of the ability to produce IL-2 [17,28], and in severely burned patients, significantly reduced activity persists for 40 days after injury [62].

Patients treated with a standard protocol experience an increase in the number of NK cells 3 weeks after burn injury. Intravenous IgG treatment increases the number of NK cells [63]. Bender et al. [64] showed that administration of polymyxin B to burn patients does not cause a decline in NK cells activity. In addition, endotoxin-rich sera from burn patients showed an inhibitory effect on NK cells [64]. An NK cell activity increase was observed in patients with major burns who were administered anti-CD3 antibody induced activated killer (CD3AK) cells [65]. Klimpeli et al. [61] reported that heat-injured patients have defective NK cell activity against neoplastic cells. It is possible that the failure of NK cell function contributes to an increased cancer rate in the burn patient population [17].

### 2.7. Complement System

The complement system is a central part of the acute phase response (APR) and has a negative impact on the local pathophysiology of a burn wound [66]. Burn wounds cause excessive activation of complement and C-reactive proteins (CRP), which increases the risk of SIRS and adversely affects the healing process of a burn wound [14,66]. Since CRP as a marker for SIRS is not practical by its sensitivity and high-level, serum procalcitonin (PCT) was suggested [67]. In patients with burns, high serum levels of CRP persist longer than PCT [68]; thus, the use of PCT allows for a more specific prediction of patient response to treatment. In addition, PCT makes it able to distinguish a patient with a septic burn from a patient with a non-septic burn [69].

Mannose-binding lectin (MBL), a protease mediating the complement lectin pathway, has been shown to play an important role in the host’s first line of defense against post-burn infectious agents through initiation of the lectin pathway [66]. MBL has been shown to be involved in the spontaneous separation of the scab formed after burn injury. This was confirmed in a study by Møller-Kristensen and co-workers [70] during which it was demonstrated that MBL-null mice had this process inhibited compared to wild-type mice. It is suggested that matrix metalloproteinase (MMP) activities that are involved in scab separation are MBL dependent. This was demonstrated by showing that MBL-null mice had reduced MMP activity in scab skin homogenates [70].

## 3. Infections

There are many factors affecting the body’s susceptibility to infections. Size and the deepness of burn injury come first among these factors [71,72]. Colonies of bacteria can be observed naturally in low concentrations—less than 10^5^ colony-forming units (CFU), without causing any infection. In the case of burn injury, physical barriers are broken, allowing pathogens access to unprotected soft tissue or circulatory areas [5,73]. Infection is a condition in which the CFU level in the wound exceeds 10^5^, and sepsis and discoloration of the burn wound may be present [5]. Because burn wounds do not contain epidermis, they are a suitable site for bacterial growth. The lack of natural protection provided by epidermal cells promotes the development of infection, and as a result, a variety of bacterial flora contaminates the wound surface within hours after the burn [74].

Among the variety of pathogens, some of them are more virulent than others. The more virulent ones release enzymes or use their flagella in order to travel deeper into the burned tissue and disperse to other soft tissues. Because of the absence of host defense, the body cannot tolerate this invasion. Then, these pathogens spread through vascular and lymphatic vessels, which results in bacteremia and sepsis [74,75,76]. The most prominent causes of sepsis and death include burn wound sepsis and various forms of infections, such as pneumonia, urinary tract infections, or central line catheter infections [72]. 

### 3.1. In the Wound 

Burn injuries provide an ideal habitat for opportunistic pathogens to grow and infect tissues [77]. Damaged microbiota and broken skin barriers enable these opportunistic microorganisms to invade the tissue and cause infection [78,79]. High quantities of bacteria indicate the presence of an infection [5].

In the early stages of an injury, Gram-positive infections are more prevalent, whereas Gram-negative bacteria infections become more dominant around 21 days after [17,70]. According to research conducted by Latifi and Karimi [75], *Staphylococcus* spp. (55.1%) are the most abundant bacteria found in burn wound cultures, followed by *Pseudomonas aeruginosa* (14.29%), *Enterococcus* sp. (12.24%), *Escherichia coli* (4%), *Klebsiella* sp. and *Proteus* sp. (both 2%). Commensals such as *Staphylococcus epidermidis* and *Propionibacterium acnes* present in the burn wound maintain homeostasis, reduce the secretion of proinflammatory cytokines [80] and negatively affect wound healing [81].

Viral burn wound infections and fungal aspergillosis or *Candida* species can also appear in the wound occasionally. The most common viruses that cause infection are *Herpes simplex* and *Varicella–zoster*. *Herpes simplex* also causes infection when the burned wound is healing [5]. 

### 3.2. Pneumonias 

Pneumonia is a severe infectious respiratory illness that causes inflammation of the pulmonary parenchyma [82]. Inhalation injury induces the risk of infection, causing lower respiratory tract diseases, especially pneumonia [83,84]. Burn-related pneumonia is linked to prolonged intubation and mechanical ventilation, and it involves up to 40% of patients who are intubated. It is the main reason for morbidity and mortality in burn patients [17,85]. Liodaki et al. [83] indicated that 22.9% of burn patients had pneumonia and 10.9% of them died while they were in Burn Care Unit [83].

Gram-negative rods, especially *P. aeruginosa*, are the major reason for burn-related pneumonia [5,74]. Bacteria such as *Streptococcus* sp. and *Haemophilus* sp. are naturally occurring organisms in normal flora of the respiratory tract, but after burn injury, they become infectious pathogens that cause pneumonia. Non-native species, such as *E. coli*, *Acinetobacter* sp., *Fusobacterium* sp. and *Peptostreptococus* sp. are among these pathogens that cause pneumonia [86].

### 3.3. In Blood and Urine 

Blood stream infections (BSI) are one of the most dangerous infections that burn patients may have. In order to indicate the BSI, there should be pathogens at least in two blood cultures, or one positive culture, of a patient with sepsis [86,87]. These pathogens, such as diphtheroids, *Bacillus* sp., *Propionibacterium* sp., coagulase negative *Staphylococci*, or *micrococci*, are not commonly considered as skin pathogens [86]. The fourth most common pathogen isolated from blood cultures in an intensive care unit is *Candida albicans*. Invasive infection with molds such as *Aspergillus* is more closely associated with mortality [5,88]. 

In predisposed burn patients, extended or inappropriate use of urinary catheters enables hematogenous dispersion of bacteria and fungi, which results in urinary tract infections [5]. In this case, urinary catheters should be removed or changed immediately. Gram-negative rods such as *Acinetobacter* spp., fungi such as *Candida* spp., and bacteria from *Enterobacteriaceae* family are the main pathogens isolated from urinary infections [5,73,89]. 

### 3.4. In the Vascular System 

Vascular access devices are required for therapy for the majority of serious burn patients. Because of wound treatment and decreased skin quality, the injury is unable to be dressed with conventional bandages; thus, central venous catheters (CVCs) must be inserted through burnt tissue. These restrictions raise the risk of central line-associated bloodstream infection (CLABSI), yet they are necessary for treatment [73]. Despite their life saving roles, they can become contaminated with pathogens such as yeast, Gram-negative bacteria or Gram-positive bacteria. Staphylococcus epidermidis and Staphylococcus aureus are the most frequent bacteria isolated from catheters [5,17]. Catheter-associated infection and septic thrombophlebitis rates in burn patients are as high as 57% [90,91]. According to the American Burn Association, a central venous catheter should be deemed as the cause of an infection if it is present at any moment in the 48 hours prior to infection, even if it has been removed in the meantime [86].

## 4. Cytokines

Innate immune cells constitute the first line of defense and are involved in recognizing initial threats. They promote the recruitment of additional immune cells through the release of cytokines that bridge the innate and acquired immune systems [92]. Cytokines are small proteins with a molecular weight of 5-20 kDa that function in inflammatory cell signaling [17]. They are produced by nearly every cell to regulate and influence immune response (Figure 3) [93,94]. They include interleukins (IL), interferons (INF), tumor necrosis factors (TNF), lymphotoxins (LT), colony-stimulating factors (CSF), chemokines and miscellaneous cytokines [95]. Burn and cell damage can lead to the release of intracellular molecules known as DAMPs [96]. PRRs are present in the host body to recognize DAMPs [97]. Released DAMPs activate the innate immune system by interacting with PRRs to produce proinflammatory cytokines [98]. These cytokines generate a variety of responses. They activate microbicidal functions of immune cells that control or eliminate infections after primary injury. Production of cytokines during severe burns can be excessive, which can cause an overabundance of inflammation. As Bidani and co-workers [99] discovered, in response to stimuli such as smoke inhalation, cytokines cause a hyperactive effect in macrophages. Several established cytokines that contribute to burn immune responses are detailed below.

### 4.1. Proinflammatory Responses 

#### 4.1.1. IL-1

The family of IL-1, through its control of many innate immune processes, is a major regulator of inflammation [100]. IL-1 has a wide range of biological functions, which include acting as a leukocytic pyrogen, a mediator of fever and a leukocytic endogenous mediator, and an inducer of several components of the acute-phase response and lymphocyte-activating factor (LAF) [101,102]. In burn injuries, increased levels of IL-1β have been reported in patients during the first week after injury [103,104]. Kupper and co-workers [104] suggest that the source of IL-1β activity is the injured keratinocyte. The release of IL-1β from the wound into the systemic circulation accounts in part for the metabolic changes that include temperature elevation, skeletal muscle proteolysis, and alterations in the production of certain serum proteins by the hepatocyte. Furthermore, epidermal IL-1β released from damaged tissues is also a potent T-cell chemoattractant; thus, it is suspected that burn wound IL-1β may affect sequestration of T cells near the burn wound, resulting in T cell lymphopenia [17]. In addition, IL-1β participates not only in extracellular matrix remodeling, but also in fibrogenesis. It is overexpressed in the early inflammatory phase of normal wound healing but is also increased in impaired wound repair and fibrosis [105,106]. This makes IL-1β one of the cytokines that has been shown to be a predictor of the presence of hypertrophic scarring after burns [17].

#### 4.1.2. IL-6

Macrophages play an important role in the initial phase of the inflammatory response, particularly mediated by their production of proinflammatory cytokines, including IL-6 [14]. It has been reported that during a burn, macrophages are hyperactive and have an increased capacity for the production of proinflammatory mediators. This was confirmed by the results of studies, where an increased level of IL-6 was detected in burn patients [48]. The highest level of IL-6 in the plasma of burn patients was within 6 hours after burn injury. The levels of IL-6 have been shown to be proportional to the size of the burn, and persistently high levels of IL-6 post burn injury may be indicative of both the severity of the burn and likelihood of mortality [106]. IL-6 is one of the most consistently elevated inflammatory mediators in burn patients, and its high levels on the first day after injury increase the risk of developing sepsis [107]. Gauglitz and co-workers [108] observed that pediatric burn patients with inhalation injury have a greater risk for mortality if IL-6 levels are increased early after injury. Qiao et al. [109] confirmed these results in trauma patients and in experimental models of sepsis, where early detection of high IL-6 is predictive of multiple organ dysfunction syndrome and mortality [109].

#### 4.1.3. TNF-α

TNF-α is one of the proinflammatory cytokines secreted by macrophages and lymphocytes in response to cell damage caused by infection or malignant transformation. It can also be generated from other type of cells and tissues, e.g., adipocytes [110]. Its main role is regulation of immune cells that generate the acute phase reaction to burn injuries. It has been shown that human TNF is able to conduct the complex orchestration of events leading to the inflammatory response [111,112]. It causes the release of IL-1 and IL-6, thus enhancing the non-specific immune response to sepsis, burn injury or inhalation injury [17]. TNF also increases vasodilation and capillary permeability by raising the level of nitric oxide in immune and endothelial cells [17]. It helps to fight systemic infections by promoting IL-1, which drives the fever response in the hypothalamus. TNF-α may play an important role in prognosis after burn injury. It was observed that impaired TNF response correlated with deceased survival in elderly burn patients [113]. Moreover, TNF-α contributes to immune hyper-responsiveness after burns and tissue destruction and increases the cachectic response through IL-1 and IL-6. It acts as the most potent catabolic cytokine. Torre-Amione et al. [114] showed that, through IL-1, it increases apoptosis in liver, muscle, and fat cells, which can lead to myocardial depression in critically ill patients.

#### 4.1.4. IFN-γ

IFN-γ, or immune IFN, is primarily produced by T lymphocytes in response to antigens or mitogens [95]. It is the earliest detectable cytokine at the site of immunization with protein antigens and plays a crucial role in the activation of innate and adaptive responses to bacteria [17,95]. IFN-γ also enhances expression of MHC class II on antigen-presenting cells, leading to more efficient antigen presentation [60]. Increased levels of IFN-γ are observed in burns initially, but they resolve over time [115]. It is unknown as of now if the immunosuppression observed after burn injury is due to reduced IFN-γ activity. In addition, decreased T-cell proliferation and impaired ability to produce IFN-γ have been observed in burn patients [116,117]. It is suspected that the reduced immune response in fighting burn infection may be related to reduced IFN-γ production [115].

### 4.2. Anti-Inflammatory Responses

#### 4.2.1. IL-10

IL-10 is known as a major immunoregulatory cytokine, mainly produced by T-cells and monocytes [118]. It has originally been called “cytokine synthesis inhibitory factor” (CSIF) because of its inhibitory effects on the release of cytokines such as IFN-γ, TNF-α, and IL-1, -6, and -8 [119]. Dehne and co-workers [119] noticed higher amounts of IL-10 in patients a short time after burn injury. However, the role of IL-10 in the initiation of immunodepression after major trauma remains controversial. Researchers reported that patients with higher levels of IL-10 were more likely to have incidence of septic events [120,121]. Thus, the immunosuppressive effects of IL-10 may be due to the suppression of acquired immune responses by IL-10 [17].

#### 4.2.2. TGF-β

Transforming growth factor-beta (TGF-β) is another well-known anti-inflammatory cytokine. It has four isoforms (TGFβ1, TGFβ2, TGFβ3, and TGFβ4), which play an important role in preventing auto-immune conditions and resolving primary immune responses. Three of them (TGF-β1, -β2, -β3) are secreted as inactive latent precursors that require activation prior to binding to the TGF-β receptors [122]. They share 60-80% homology and appear to be present in wound healing, including in wounds from early human fetuses, which repair cutaneous wounds perfectly [123,124]. However, isoforms in fetal wounds differ in levels of expression, duration in the wound and biological activity [125]. Wang and co-workers [126] suggested a possible role for TGF-β1 in hypertrophic scar formation. They showed that hypertrophic-derived fibroblasts and hypertrophic scar tissue produced more mRNA and proteins for TGF-β1 than normal skin or fibroblasts derived from normal skin. Moreover, it was also demonstrated that hypertrophic-derived fibroblasts have increased expression of TGF-β1 receptors compared with normal skin [127]. Schmid and co-workers [127] suggested that this expression of TGF-β1 is persistent compared to normal wound healing. Most likely, this persistence of receptor expression may cause a fibrotic phenotype. In addition, TGF-β1 has been shown to modulate immune function in the body, which causes an antiproliferative effect on B and T lymphocytes. It inhibits the transition of B cells from the G1 phase to the S phase of the cell cycle, which is the reason for the inhibition of proliferation [128]. TGF also causes a class change in antibodies to IgA and an overall reduction in the secretion of most Ig by inhibiting mRNA induction [129]. Ishikawa et al. [130] showed that TGF-β1 is involved in the suppression of humoral immunity after burn injury. Their results show significantly higher TGF mRNA expression in the spleen and liver of mice after burn injury. Additionally, they showed that the impaired production of IgM and IgG is caused by TGF’s inhibition of the proliferation of cells secreting these immunoglobulins. However, TGF-β1 does not impair the production of IgM and IgG by cells. 

## 5. Antioxidant and Trace Element Supplementation

Treatment of burns is difficult because it must affect both the intensity of the proinflammatory and anti-inflammatory response [2]. However, the main goal of treatment is to reduce inflammation and improve patient prognosis. Numerous studies on this topic have already contributed to reduced mortality and more effective stabilization of patients, but the topic still faces difficulties. As of today, therapies that focus on specifically blocking cytokines such as IL-1, TNF or IL-6 have been found to be ineffective [42]. For this reason, the focus has been on other strategies to reduce the inflammation and hypermetabolism associated with burn injury. One option to improve clinical outcomes is supplementation with antioxidants (e.g., ascorbic acid, glutathione, acetylcysteine, or vitamins A, E, and C) and trace elements [39]. The use of this strategy scavenges free radicals, inhibits the formation of new ones, and through these actions, reduces inflammation and hypermetabolism in patients after burn injury [51]. Antioxidants are a key component of mechanisms that reduce the formation of reactive oxygen species and attenuate the synthesis of transcription factors such as NF-κB contributing to reduced inflammation [52]. Literature reports indicate not only the positive effects of antioxidants on the survival of burn patients, but also on faster wound healing [131,132]. A study by Rehou et al. [42] showed that burn patients had better outcomes after supplementation with antioxidants and trace elements. Patients had reduced levels of cytokines such as IL-1β and IFN-γ, but there were no differences in IL-6 levels. Caffeic acid phenethyl ester (CAPE) has also shown strong antioxidant properties. As an anti-inflammatory, antiviral, antimicrobial, and immunostimulant, it is effective in the treatment of burn injury, which was confirmed in rat models [133]. Another interesting compound with antioxidant properties is melatonin, which not only contributes to the reduction of oxidative stress but also shows protective effects in burn-induced skin damage and appears to be a promising agent for inhibiting the activation of blood coagulation after severe burns [134,135].

## 6. Conclusions

Innate immunity plays a key role after burn injury. When physical barriers are broken, pathogens have access to unprotected soft tissue or circulatory areas. In this case, cells of the innate immune system are an essential part of the organism’s protection. Proinflammatory cytokines, produced by cells, activate microbicidal functions that control or eliminate infections after primary injury. However, the inflammatory responses are coupled to counter regulatory mechanisms, and in severe burn injuries, patients can easily be overwhelmed by both anti-inflammatory and inflammatory mechanisms. During severe burns, patients are also at risk of secondary infections because wounds are suitable for bacterial growth. Such phenomena are dangerous and can cause immunosuppression. The acquired knowledge of the immune system, nonspecific innate mechanisms, protein signaling networks of complement, coagulation and cytokines have contributed to the increased survival of burn patients over several decades. In addition, research into the therapeutic options for patients after burn injury is contributing to improved clinical outcomes. Antioxidants that reduce inflammation and hypermetabolism appear to be a promising method to reduce patient mortality. However, not all mechanisms occurring within the body of individuals with burn injuries have been understood or explained thus far. Further research and their elucidation may provide the most effective tools to optimize immune function after burn injury.

## Figures and Tables

**Figure 1 ijms-23-00716-f001:**
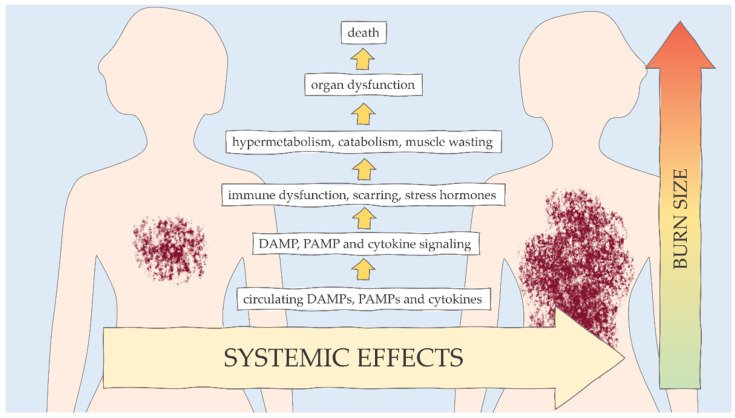
Systemic responses are dependent on the size of the burn (based on [1,2,3,4]).

**Figure 2 ijms-23-00716-f002:**
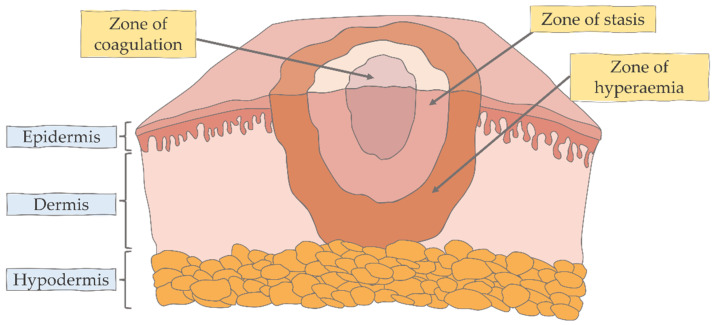
Jackson’s three zones in burns.

**Figure 3 ijms-23-00716-f003:**
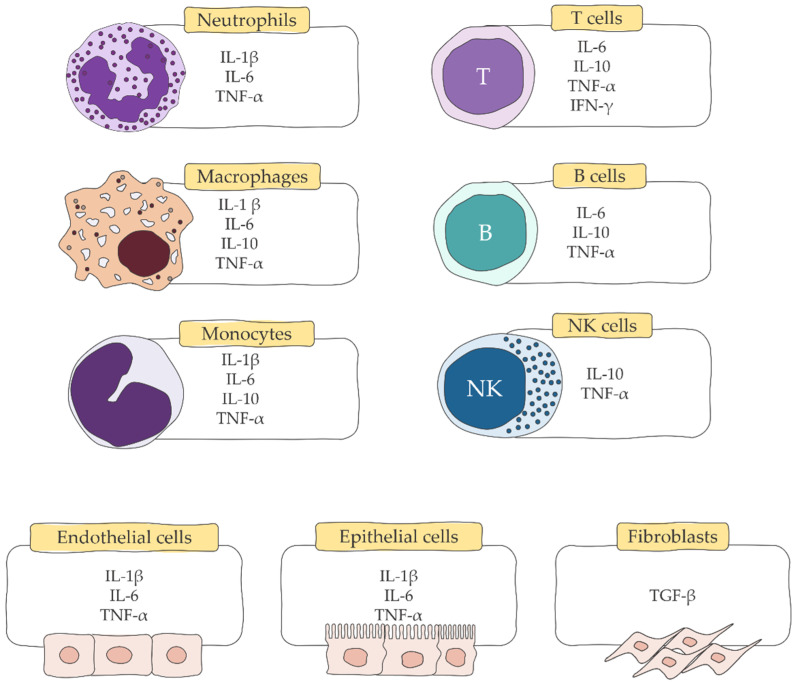
Schematic representation of cells involved in the immune response expressing different cytokines. Abbreviations: IL, interleukin; TNF, tumor necrosis factor; IFN, interferon; TGF, transforming growth factor; NK, natural killer (based on [93,94]).

## Data Availability

Not applicable.

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
