# Peer review of "Innate Immune System Response to Burn Damage—Focus on Cytokine Alteration"

_ijms, 2022, doi:10.3390/ijms23020716_

Round 1

Reviewer 1 Report

In this paper, the immune response after burns is reviewed with respect to immune cells, cytokines and infection. The manuscript is however highly out of balance and does not cover all major aspects.

Although burn patients are at high risk of attracting microorganisms in their wounds, which might lead to colonization and ultimately infection, infection is not commonplace in burns. On various occasions, the authors seem to regard the reaction towards pathogens/antigens as the prime immune response (and not so much towards the thermal injury). For example “host immune response” in the title and section 2 suggests this. This aspect (section 4) should be removed OR be the main topic.

The authors attempt to cover all aspects, but in doing so neglect several aspects or provide only very limited details. Various subjects are missing: phagocytosis, migratory potential, thrombocytes, B cells, several cytokines such as MCP1 and IL8. On the other hand, more attention is being paid to less relevant cells in burns such as NK, NKT and a long list of lymphocyte subtypes, which are not further discussed.

Minor issues

What is the purpose of the last paragraph section 2.1?

After burns, a major proportion of neutrophils is immature, which affects their function. Please discuss.

The high neutrophil count over the reduced lymphocytes (neutrophil lymphocyte ratio or NLR) can be used as an indicator for sepsis.

2.3 DC only secrete cytokines “when infection is diagnosed”?

It is more informative to characterize monocytes as classical, alternative and intermediate. M1 and M2 labels are often used to label macrophages but it is becoming more and more clear that this differentiation is too simple. Section 2.4 and 2.5 do not provide enough info/detail/background.

2.6 T cells “Patients with oppression have had changes in lymphocyte response, suppression [45,47] and decreased gene expression”. What is meant here? The second paragraph lacks structure.

2.6 (?) inflammasomes and 2.7 complement. Why put these components in part 2? In addition, there is not enough info/detail. Why not discuss the use of procalcitonine instead of CRP as sepsis parameter? CRP is always high in most burn patients.

Overall grammar and spelling should be improved.

Figure 4 is not very informative and too simplistic.

The interleukin 1 family contains 11 different cytokines. I assume you refer to IL1beta?

TNFα acts as a growth factor for fibroblast angiogenesis, please explain.

IFNγ is the earliest detectable cytokine at the site of immunization, but this does not match the topic of the manuscript, which is the immune response after burns. The lower ability to immunzie and produce antibodies against invading microorganisms is presently not included.

The different types of TGFβ have different dynamics and roles. They are indeed import in wound healing, where they induce for example differentiation of fibroblasts (for the production of collagen for example). What is the effect TGFβ on immune cells in burns?

I would suggest to remove the entire section 4.

105 CFU should be 105 (100,000) CFU, which is not a naturally low concentration!

Why mention a relation between epidermis and heat? Without epidermis, the wound is warmer and heat is lost, but there is also no shielding the tissue > availability of nutrients, moisture, no defense (AMPs, etc). “Burn wound itself is the primary cause of burn infections” is incorrect.

A definition for infection should be included, bacterial presence is not the only criterium in burns.

Pathogens are not the only microorganisms in burn wounds. Commensals can also colonize wounds and cause complications.

Author Response

Dear Reviewer,

Thank you for giving us the opportunity to submit a revised draft of our manuscript titled: “Immune system response to burn damage, focus on cytokines alterations and infections“ to International Journal of Molecular Sciences. We appreciate the time and effort that you have dedicated to providing your valuable feedback on our manuscript. We are grateful to you for your insightful comments on our paper. We have been able to incorporate changes to reflect most of the suggestions. We have highlighted the changes within the manuscript.

Here is a point-by-point response to the reviewer’s comments and concerns.

Reviewer 1

In this paper, the immune response after burns is reviewed with respect to immune cells, cytokines and infection. The manuscript is however highly out of balance and does not cover all major aspects.

Although burn patients are at high risk of attracting microorganisms in their wounds, which might lead to colonization and ultimately infection, infection is not commonplace in burns. On various occasions, the authors seem to regard the reaction towards pathogens/antigens as the prime immune response (and not so much towards the thermal injury). For example “host immune response” in the title and section 2 suggests this. This aspect (section 4) should be removed OR be the main topic.

The authors attempt to cover all aspects, but in doing so neglect several aspects or provide only very limited details. Various subjects are missing: phagocytosis, migratory potential, thrombocytes, B cells, several cytokines such as MCP1 and IL8. On the other hand, more attention is being paid to less relevant cells in burns such as NK, NKT and a long list of lymphocyte subtypes, which are not further discussed.

Minor issues

  • What is the purpose of the last paragraph section 2.1? After burns, a major proportion of neutrophils is immature, which affects their function. Please discuss. The high neutrophil count over the reduced lymphocytes (neutrophil lymphocyte ratio or NLR) can be used as an indicator for sepsis.

Thank you for bringing this to our attention. After discussion, we have decided to remove this paragraph.

  • 3 DC only secrete cytokines “when infection is diagnosed”?

The text has been corrected to “and when an infection occurs”.

  • It is more informative to characterize monocytes as classical, alternative and intermediate. M1 and M2 labels are often used to label macrophages but it is becoming more and more clear that this differentiation is too simple.

The text has been corrected. The suggested division has been provided as per your feedback.

  • Section 2.4 and 2.5 do not provide enough info/detail/background.

Section 2.4 and 2.5 has been supplemented with additional information such as an updated division of macrophages and a more detailed description of the role of mast cells in burns.

  • 6 T cells “Patients with oppression have had changes in lymphocyte response, suppression [45,47] and decreased gene expression”. What is meant here? The second paragraph lacks structure.

The sentence has been grammatically corrected to “Changes in lymphocyte response, suppression and decreased lymphocyte gene expression were observed in burn patients.”

  • 6 (?) inflammasomes and 2.7 complement. Why put these components in part 2? In addition, there is not enough info/detail. Why not discuss the use of procalcitonine instead of CRP as sepsis parameter? CRP is always high in most burn patients.

Thank you for this suggestion. We putted it here because we wanted to highlight their role in innate immune response and because it is very under-researched topic and we would like to encourage research on this. Details regarding complement, inflammasome, and procalcitonine were completed.

  • Overall grammar and spelling should be improved.

The text has been checked for linguistic correctness.

  • Figure 4 is not very informative and too simplistic.

After reviewing the opinion, we decided to remove this figure from our manuscript.

  • The interleukin 1 family contains 11 different cytokines. I assume you refer to IL1beta?

Yes, the text has been corrected. We have indicated in the text where the entire IL-1 family is described and where IL-1beta.

  • TNFα acts as a growth factor for fibroblast angiogenesis, please explain.

The text has been corrected. TNF is a factor that causes lymphocyte recruitment. Lymphocytes act as a growth factor for fibroblast angiogenesis. After discussion, we felt this was redundant information that adds nothing to the review, so the excerpt "During the chronic phase of wound healing, lymphocytes are recruited by TNF to clean-up of wound debris and act as a growth factor for fibroblast angiogenesis." has been removed.

  • IFNγ is the earliest detectable cytokine at the site of immunization, but this does not match the topic of the manuscript, which is the immune response after burns. The lower ability to immunzie and produce antibodies against invading microorganisms is presently not included.

Thank you for your feedback. IFN-É£ was presented by us because it is a cytokine whose effects during burn injury have not been fully elucidated. We wanted to highlight the fact that its reduced production may be a factor in the more severe control of infections during burns that we also mention in the publication.

  • The different types of TGFβ have different dynamics and roles. They are indeed import in wound healing, where they induce for example differentiation of fibroblasts (for the production of collagen for example). What is the effect TGFβ on immune cells in burns?

The text has been corrected. We explained effect TGFβ on immune cells in burns. It causes an antiproliferative effect on B and T lymphocytes. It inhibits the transition of B cells from the G1 phase to the S phase of the cell cycle which is the reason for the inhibition of proliferation. TGF also causes a class change in antibodies to IgA and an overall reduction in the secretion of most Ig by inhibiting mRNA induction.

  • I would suggest to remove the entire section 4.

We decided to leave Section 4 in the manuscript. However, according to your suggestion, we have highlighted the importance of the infection topic by reordering the sections and changing the manuscript title to "Immune system response to burn damage, focus on cytokine alterations and infections". Thank you for this recommendation.

  • 105 CFU should be 105 (100,000) CFU, which is not a naturally low concentration!

The error has been corrected: “Colonies of bacteria can be observed naturally in low concentrations, less than 105 colony-forming units (CFU), without causing any infection.”. Thank you for bringing this important issue to our attention.

  • Why mention a relation between epidermis and heat? Without epidermis, the wound is warmer and heat is lost, but there is also no shielding the tissue > availability of nutrients, moisture, no defense (AMPs, etc). “Burn wound itself is the primary cause of burn infections” is incorrect.

The text has been corrected. We have removed the incorrect statement and added an important comment about the lack of sufficient protection normally offered by the epidermis.

  • A definition for infection should be included, bacterial presence is not the only criterium in burns.

The definition has been added: “Infection is a condition in which the CFU level in the wound exceeds 105, and sepsis and discoloration of the burn wound may be present”

  • Pathogens are not the only microorganisms in burn wounds. Commensals can also colonize wounds and cause complications.

Information about commensals has been added, "The commensals like Staphylococcus epidermidis and Propionibacterium acnes present in the burn wound maintain homeostasis, reduce the secretion of pro-inflammatory cytokines and negatively affect wound healing”

Once again thank you for your time and consideration. We do hope that in the corrected form the paper will fulfil the requirements.

On behalf of the Authors,

Paulina Niedźwiedzka-Rystwej

Reviewer 2 Report

Manuscript ID: ijms-1472941

The review entitled “The role of the immune system in burns - host immune response, cytokine alterations and infections” by Sierawska et al. perfectly explained the role of the inflammatory and immune mediators that are activated after a burn event. The authors fully described the production, release, and action of the immune mediators. Furthermore, the failure of the immune systems again microorganisms infection in different cases and parts of the body was usefully discussed and updated to the most recent literature on the topic.

I would like to suggest an improvement for review on the part of the inflammatory and immune mediators, asking if the use of antioxidant or anti-inflammatory could be a problem or not and trying to figure out may a limit of when and where they should be applied during burn management. This because completely switching off the inflammation as well as the presence of free radicals could block the next steps of the healing. On the other hand, in the case of severe burns may be useful to reduce the excessive inflammation and production of free radicals.

After this minor comment, I believe the manuscript is ready for publication.  

Author Response

Dear Reviewer,

Thank you for giving us the opportunity to submit a revised draft of our manuscript titled: “Immune system response to burn damage, focus on cytokines alterations and infections“ to International Journal of Molecular Sciences. We appreciate the time and effort that you have dedicated to providing your valuable feedback on our manuscript. We have been able to incorporate changes to reflect most of the suggestions. We have highlighted the changes within the manuscript.

Here is a point-by-point response to the reviewer's comments and concerns.

The review entitled “The role of the immune system in burns - host immune response, cytokine alterations and infections” by Sierawska et al. perfectly explained the role of the inflammatory and immune mediators that are activated after a burn event. The authors fully described the production, release, and action of the immune mediators. Furthermore, the failure of the immune systems again microorganisms infection in different cases and parts of the body was usefully discussed and updated to the most recent literature on the topic.

I would like to suggest an improvement for review on the part of the inflammatory and immune mediators, asking if the use of antioxidant or anti-inflammatory could be a problem or not and trying to figure out may a limit of when and where they should be applied during burn management. This because completely switching off the inflammation as well as the presence of free radicals could block the next steps of the healing. On the other hand, in the case of severe burns may be useful to reduce the excessive inflammation and production of free radicals.

After this minor comment, I believe the manuscript is ready for publication. 

Thank you for your feedback. We are very pleased that you like the publication. As advised, we have added a chapter on the use of antioxidants in the treatment of patients with burn injuries. In Chapter 5, we described how antioxidants help reduce inflammation and hypermetabolism and patients had better outcomes after supplementation with antioxidants and trace elements.

Once again thank you for your time and consideration. We do hope that in the corrected form the paper will fulfil the requirements.

On behalf of the Authors,

Paulina Niedźwiedzka-Rystwej

Round 2

Reviewer 1 Report

The manuscript has been improved substantially. Excluding infection and restructuring have been positive. “5. Antioxidant and Trace Element Supplementation” is a nice addition. But the paper still requires some amendments.

Title: infections are no longer a major subject, “and infections” can be removed from the title.

Abstract requires some fine tuning: “burns stimulate the immune system but result in higher susceptibility”? this needs some explanation. And only a properly functioning immune system is effective. This implies that this is not the case in burn patients. Immunosuppression: of which aspects? Typo: co-infections, “co-“ can be omitted.

Page 3: “coagulation of component proteins”, remove component. “despite the presence of sterility”  should be rephrased.

2. phagocytes also encompasse macrophages and neutrophils.

2. Burns generally result in a hyperactive immune response. Where does immunosuppressive phase relate to?

2.1 according to the response, this section was supposed to be removed. The first cells to respond to burns are (thrombocytes and) mast cells and neutrophils. NK and NKT play a role in viral infections and on tumors. Their role in burns is much less important than suggested here. This part is not in balance compared to the role of macrophages, neutrophils and mast cells.

2.2 Effect of burns on phagocytosis and migration is still not included, why not?

2.4 this section needs much more information and explanation, it raises various questions. Monocytes and macrophages are of major importance in burns.

2.5 Overall this section requires more interpretation and grammar should be improved. What is the definition of inflammasomes? How is NLRP3 expression associated with mortality? What does ”NP3R has the capability to identify DAMPs as the uniquely inflammasome” mean, what are the consequences? What can we conclude from “deficiency in NLRP3 expression lead to increased chemotaxis and macrophage activity”? “Bacterial endotoxins disrupt the intestinal epithelial barriers”? Is that the case or is permeability increased resulting in translocation of bacteria from the intestinal tract to the blood stream?

2.6 There is much more literature on mast cells in burns, wound healing and scar formation than presented here. The grammar needs improvement. Typo rod > rodent.

2.7 typos CPR > CRP. What is the point of mentioning MBL, without further discussion?

2.8 the relevance of these subsets of lymphocytes in burns is still not made clear. CD8 T cells are cytoxic T cells, T exhausted and T anergic are subsets or cell status? More details are needed in first half of page 7. What kind of changes, what was suppressed/decreased response, etc. Early excision leads to the restoration of T cell responses: how?

Figure 3: using IL-1 here does not seem correct, should be specified.

4.2.2 Specify the TGF type (TGF-β1?)

5. interesting addition, but grammar can be improved. For example “The use of this strategy scavenges free radicals, inhibits the formation of new ones, and thus reduces inflammation”

Author Response

Thank you again for your kindness and consideration of our manuscript. We analysed the paper again and tried to correct the mistakes that the Reviewer came across.

We would like to add, that in the version sent out for re-review, the section on infections was mistakenly removed, but we would like to leave it in our paper. The purpose of our work is to show that despite the very active participation and rapid response of the immune system, infections can still occur. This cause and effect sequence is an important element in the paper for us. We hope you understand that, but on the same time we would like to assure that we are ready to make any further amendments needed.

Below are point-by-point answers to the Reviewer concerns.

Title: infections are no longer a major subject, “and infections” can be removed from the title.

Thank you for your comment, the suggested part of the topic has been removed. Infections still appear in the paper, just not as a major topic, so they do not need to be highlighted in the text.

Abstract requires some fine tuning: “burns stimulate the immune system but result in higher susceptibility”? this needs some explanation. And only a properly functioning immune system is effective. This implies that this is not the case in burn patients. Immunosuppression: of which aspects? Typo: co-infections, “co-“ can be omitted.

Compliant recommendations have been added to the abstract: “As a result of the loss of the protective ability of the epidermis, microbes including bacteria, fungi, and viruses have easier access to the system and can result in infections. However, the patient is still able to overcome the infections that occur through a cascade of cytokines and growth factors stimulated by inflammation.” and“This immunosuppression includes apoptosis induced lymphopenia, decreased IL-2 secretion by T lymphocytes, neutrophil storm, impaired phagocytosis, and decreased monocyte human leukocyte antigen-DR.”, and the typo was removed.

Page 3: “coagulation of component proteins”, remove component. “despite the presence of sterility”  should be rephrased.

“Component“ has been removed, and the specified fragment rephrased to "and even though it seems that the burn wound should be free of microorganisms because of the high temperature,”

  1. phagocytes also encompasse macrophages and neutrophils.

This has been included.

  1. Burns generally result in a hyperactive immune response. Where does immunosuppressive phase relate to?

The sentence has been corrected to “which results in a strong inflammatory response in the immune system, defined as systemic inflammatory response syndrome (SIRS)”

2.1 according to the response, this section was supposed to be removed. The first cells to respond to burns are (thrombocytes and) mast cells and neutrophils. NK and NKT play a role in viral infections and on tumors. Their role in burns is much less important than suggested here. This part is not in balance compared to the role of macrophages, neutrophils and mast cells.

Our apologies, the last paragraph was not removed by mistake. It no longer appears in the current version. The order and data about cell response to burns have been corrected and the order of sections has been changed as suggested. To emphasize the role of mast cells, they have been moved to the first section.

2.2 Effect of burns on phagocytosis and migration is still not included, why not?

We have included the effect of burns on phagocytosis and migration (chemotaxis) in the section provided.

2.4 this section needs much more information and explanation, it raises various questions. Monocytes and macrophages are of major importance in burns.

Details have been added and the migration has been completed.

2.5 Overall this section requires more interpretation and grammar should be improved. What is the definition of inflammasomes? How is NLRP3 expression associated with mortality? What does ”NP3R has the capability to identify DAMPs as the uniquely inflammasome” mean, what are the consequences? What can we conclude from “deficiency in NLRP3 expression lead to increased chemotaxis and macrophage activity”? “Bacterial endotoxins disrupt the intestinal epithelial barriers”? Is that the case or is permeability increased resulting in translocation of bacteria from the intestinal tract to the blood stream?

Thank you for your feedback. We have corrected the definition of inflamasome and further described how NLRP3 expression affects patient mortality by causing stress-induced diabetes. The sentence "deficiency in NLRP3 expression lead to increased chemotaxis and macrophage activity" was incorrect and has been corrected to "deficiency in NLRP3 expression was observed, leading to a decrease in the expression of factors involved in wound healing processes. Reduced production of proinflammatory cytokines and chemokines impairs not only keratinocyte migration and proliferation, but also chemotaxis of immune cells"

2.6 There is much more literature on mast cells in burns, wound healing and scar formation than presented here. The grammar needs improvement. Typo rod > rodent.

Thank you for your opinion. The error regarding the word "rodent" and the grammar has been corrected. Regarding the bibliography, we know that there is a lot of literature, but we presented the most important aspects in the paper and we didn't want it to be too long, and we already have a lot of literature (>130).

2.7 typos CPR > CRP. What is the point of mentioning MBL, without further discussion?

The CRP typo has been corrected. The topic of MBLs has been expanded to include information on the contribution of MBLs to the separation of crusts resulting from burn injury.

2.8 the relevance of these subsets of lymphocytes in burns is still not made clear. CD8 T cells are cytoxic T cells, T exhausted and T anergic are subsets or cell status? More details are needed in first half of page 7. What kind of changes, what was suppressed/decreased response, etc. Early excision leads to the restoration of T cell responses: how?

The division of lymphocytes is presented to facilitate understanding of the rest of the text, where we refer to specific lymphocytes. We feel that providing more detail to an already detailed description misses the point and topic of the paper. The sentence about changes was poorly worded and we have already corrected it. We also added a clarification to the last sentence "Based on this phenomenon, it is suggested that immunosuppressive modulators are pre-sent in this tissue, but the mechanism has not been fully elucidated"

Figure 3: using IL-1 here does not seem correct, should be specified.

The scheme has been corrected

4.2.2 Specify the TGF type (TGF-β1?)

TGF type has been added

  1. interesting addition, but grammar can be improved. For example “The use of this strategy scavenges free radicals, inhibits the formation of new ones, and thus reduces inflammation”

Thanks for your feedback and we are glad you liked the additional chapter. As advised, we have improved the gramma

Thank you again for your time and consideration.

Kind regards,

Paulina Niedźwiedzka-Rystwej

Round 3

Reviewer 1 Report

The manuscript has been improved substantially. Excluding infection and restructuring have been positive. “5. Antioxidant and Trace Element Supplementation” is a nice addition. But the paper still requires some amendments.

Title: infections are no longer a major subject, “and infections” can be removed from the title.

Abstract requires some fine tuning: “burns stimulate the immune system but result in higher susceptibility”? this needs some explanation/attentuation. And “only a properly functioning immune system is effective”. This implies that this is not the case in burn patients. Immunosuppression: of which aspects? Typo: co-infections, “co-“ can be omitted.

Page 3: “coagulation of component proteins”, remove component. “despite the presence of sterility”  should be rephrased.

2. phagocytes also encompasses macrophages and neutrophils.

3. Burns generally result in a hyperactive immune response. Where does “immunosuppressive phase” relate to?

2.1 according to the response, this section was supposed to be removed. The first cells to respond to burns are (thrombocytes and) mast cells and neutrophils. NK and NKT play a role in viral infections and on tumors. Their role in burns is much less important than suggested here. This part is not in balance compared to the role of macrophages, neutrophils and mast cells.

2.2 Effect of burns on phagocytosis and migration is still not included, why not?

2.4 this section needs much more explanation, it raises various questions. Monocytes and macrophages are of major importance in burns and deserve a larger section.

2.5 Overall this section requires more interpretation and grammar should be improved. What is the definition of inflammasomes? How is NLRP3 expression associated with mortality? What does ”NP3R (typo?) has the capability to identify DAMPs as the uniquely inflammasome” mean, what are the consequences? What can we conclude from “deficiency in NLRP3 expression lead to increased chemotaxis and macrophage activity”? “Bacterial endotoxins disrupt the intestinal epithelial barriers”? Is that the case or is permeability increased resulting in translocation of bacteria from the intestinal tract to the blood stream?

2.6 There is much more literature on mast cells in burns, wound healing and scar formation than presented here. The grammar needs improvement. Typo rod > rodent.

2.7 typos CPR > CRP. What is the point of mentioning MBL, without further discussion?

2.8 the relevance of these subsets of lymphocytes in burns is still not made clear. CD8 T cells are cytoxic T cells, T exhausted and T anergic are subsets or cell status? More details are needed in first half of page 7. What kind of changes, what was suppressed/decreased response, etc. Early excision leads to the restoration of T cell responses: how?

Figure 3: using IL-1 here does not seem correct, should be specified.

Cytokines such as MCP1 and IL8 are still lacking

4.2.2 Specify the TGF type (TGF-β1?)

5. interesting addition, but grammar can be improved. For example “The use of this strategy scavenges free radicals, inhibits the formation of new ones, and thus reduces inflammation”?

Author Response

(The authors gave the same response as above.)
